

# Evidence for an oncogenic role of HOXC6 in human non-small cell lung cancer

Yingcheng Yang[*], Xiaoping Tang[*], Xueqin Song, Li Tang, Yong Cao, Xu Liu, Xiaoyan Wang, Yan Li, Minglan Yu, Haisu Wan and Feng Chen

Experimental Medicine Center, The Affiliated Hospital of Southwest Medical University, Luzhou, Sichuan, China

[*] These authors contributed equally to this work.

## ABSTRACT

**Background**. Identification of specific biomarkers is important for the diagnosis and treatment of non-small cell lung cancer (NSCLC). HOXC6 is a homeodomain-containing transcription factor that is highly expressed in several human cancers; however, its role in NSCLC remains unknown.

**Methods**. The expression and protein levels of *HOXC6* were assessed in NSCLC tissue samples by Quantitative real-time PCR (qRT-PCR) and immunohistochemistry, respectively. *HOXC6* was transfected into the NSCLC cell lines A549 and PC9, and used to investigate its effect on proliferation, migration, and invasion using CFSE, wound healing, and Matrigel invasion assays. Next-generation sequencing was also used to identify downstream targets of HOXC6 and to gain insights into the molecular mechanisms underlying its biological function.

**Results**. *HOXC6* expression was significantly increased in 66.6% (20/30) of NSCLC tumor samples in comparison to normal controls. HOXC6 promoted proliferation, migration, and invasion of NSCLC cells *in vitro*. RNA-seq analysis demonstrated the upregulation of 310 and 112 genes in A549-HOXC6 and PC9-HOXC6 cells, respectively, and the downregulation of 665 and 385 genes in A549-HOXC6 and PC9-HOXC6 cells, respectively. HOXC6 was also found to regulate the expression of genes such as *CEACAM6, SPARC, WNT6, CST1, MMP2,* and *KRT13*, which have documented pro-tumorigenic functions.

**Discussion**. *HOXC6* is highly expressed in NSCLC, and it may enhance lung cancer progression by regulating the expression of pro-tumorigenic genes involved in pro-liferation, migration, and invasion. Our study highlighted the oncogenic potential of *HOXC6*, and suggests that it may be a novel biomarker for the diagnosis and treatment of NSCLC.

Corresponding authors
Haisu Wan, whssyzx@swmu.edu.cn
Feng Chen, chenf6305@foxmail.com

## INTRODUCTION

Lung cancer has the highest cancer incidence in the world (*Chen et al., 2016b*). Non-small cell lung cancer (NSCLC) accounts for 80–85% of the total number of lung cancer cases (*Lee et al., 2013*; *Zienolddiny & Skaug, 2011*). In spite of improvements in imaging science,

radical surgical resection, and NSCLC auxiliary detection technologies, most patients have advanced disease at diagnosis, and the overall 5-year survival rate is less than 15% (*National Lung Screening Trial Research T et al., 2011*; *Ye & Zhao, 2016*). Therefore, more effective molecular makers are needed for the diagnosis and treatment of NSCLC.

*HOX* genes belong to the homeobox gene superfamily (*Cillo et al., 2001*). The human *HOX* gene family is made up of 39 members in four clusters (A–D) located on chromosomes 7, 17, 12, and 2, with each cluster containing 9 to 13 loci (*Apiou et al., 1996*). Many *HOX* genes have been found to be closely associated with the progression of cancer (*Shah & Sukumar, 2010*). *HOXA10* can inhibit the invasion of breast cancer cells by enhancing the expression of *TP53* (*Chu, Selam & Taylor, 2004*). *HOXB7* can also promote tumor growth by upregulating the expression of angiogenic growth factors (*Care et al., 2001*). HOXC6 is a transcription factor that regulates cell differentiation during embryonic development (*Maroulakou & Spyropoulos, 2003*). Aberrant expression of *HOXC6* may result in the malignant transformation of normal cells (*DeInnocentes et al., 2015*; *Feng et al., 2009*; *Moon et al., 2012*; *Wright et al., 1989*; *Zhang et al., 2013*), and elevated *HOXC6* expression has been observed in several types of cancers, including prostate, gastrointestinal, colorectal, and hepatocellular cancers (*Chen et al., 2016a*; *Ji et al., 2016*; *Sui et al., 2016*; *Vinarskaja et al., 2011*). However, the biological function of *HOXC6* has not been well understood. Here, we report that *HOXC6* is highly expressed in NSCLC cells, and overexpression of *HOXC6* promotes the proliferation, migration, and invasion of NSCLC cells. The phenotypic effects of HOXC6 may be mediated by genes that have been previously reported to be involved in the progression of cancer. Our data also suggest that HOXC6 is a potential molecular marker for the diagnosis and treatment of NSCLC.

## MATERIALS & METHODS

### Cell lines
NSCLC cell lines A549 and PC9 were obtained from the Stem Cell Bank of the Chinese Academy of Sciences. A549 was maintained in RPMI 1640 Medium (Gibco, Thermo Fisher Scientific, Waltham, MA, USA). PC9 and 293FT cells were maintained in Dulbecco's Modified Eagle's Medium (DMEM) (Gibco, Thermo Fisher Scientific, Waltham, MA, USA), supplemented with 10% fetal bovine serum (Gibco, Thermo Fisher Scientific, Waltham, MA, USA). All of the cell lines were housed in 37 °C incubators with 5% $CO_2$ saturation.

### Human clinical specimens
Clinical specimens were collected from patients at the Affiliated Hospital of Southwest Medical University (Luzhou, China). Tissue samples were surgically retrieved from NSCLC patients after obtaining written consent and with the approval of the ethics committee of the Affiliated Hospital of Southwest Medical University (k2018003-r). Clinical samples were immediately separated into lung tumor tissues (T) and adjacent non-tumor lung tissues (N), flash frozen in liquid nitrogen, and then stored at −80 °C until further analysis.

## Immunohistochemistry (IHC)

Formalin-fixed paraffin-embedded sections were treated with 3% $H_2O_2$ for 10 min after deparaffinization in xylene and rehydration in decreasing concentrations of ethanol from 100 to 75%. Antigen retrieval was performed by using heated sodium citrate. Sections were then blocked using 10% normal goat serum to prevent non-specific antibody reactions. The sections were then incubated overnight at 4 °C, with a mouse monoclonal antibody against human HOXC6 (dilution, 1:200; Santa Cruz Biotechnology, Santa Cruz, CA, USA) After incubation, the HOXC6 antigen-antibody reaction was performed using an immunoperxodase-based kit (ZSGB Bio, Beijing, China). To quantify the expression of HOXC6, two specialist pathologists scored the stained samples based on the product of the intensity and degree of staining (0–100%). The staining intensity was categorized as follows: 0 (negative), 1 (weak), 2 (moderate), and 3 (strong). Only cytoplasmic staining was counted as positive staining.

## RNA isolation and Quantitative Real-Time Polymerase Chain Reaction (qRT-PCR)

Total RNA was extracted from NSCLC cell lines and human samples using TRIzol (TaKaRa Bio, Otsu, Japan). Then, 0.5 µg total RNA was reverse transcribed into cDNA using a PrimeScript™ RT reagent Kit with gDNA Eraser (TaKaRa Bio, Otsu, Japan) according to the manufacturer's instructions. qRT-PCR was performed using SYBR® Premix Ex Taq™ II (TaKaRa Bio, Otsu, Japan) system on an ABI QuantStudio™ 7 Flex Real-Time PCR (Applied Biosystems, USA). For the PCR reaction, specific primers were selected based on PrimerBank HOXC6 sequences. The following primers were synthesized by Invitrogen: human HOXC6 forward primer 5′-ACAGACCTCAATCGCTCAGGA-3′, and reverse primer 5′-AGGGGTAAATCTGGATACTGGC-3′; GAPDH forward primer 5′-ATGCTGGCGCTGAGTACGTC-3′, and reverse primer 5′-GGTCATGAGTCCTTCC-ACGATA-3′. Gene expression levels were normalized to GAPDH using the $2^{-\Delta\Delta Ct}$ method.

## Lentivirus packaging

The lentiviral *HOXC6-* expressing vector pCDH-*HOXC6* and empty control vector pCDH-NEO were obtained from Experimental Medicine Center, The Affiliated Hospital of Southwest Medical University. The 293FT cells were used for lentiviral packaging. A total of three vectors (pCDH-*HOXC6* or pCDH-NEO, psPAX2 and pMD2G) were co-transfected into 293FT cells using Lipofectamine3000 (Invitrogen, Carlsbad, CA, USA). After 72 h, viral supernatant was collected and stored at −80 °C.

## Cell transfection

A549 and PC9 cells were plated into a 6-well plate. Cells were then transfected with the lentiviral plasmids at 80% confluence. After 8 h, the transfection medium was replaced with complete growth medium supplemented with 500 µg/ml neomycin. qRT-PCR and Western blot were performed to evaluate transfection efficiency.

## Immunoblotting

Cells were lysed on ice using RIPA buffer (Cell Signaling Technology, USA). Protein concentration was measured by BCA kit (Beyotime, Shanghai, China). Equal amounts

of total protein were resolved using 12% SDS-PAGE gel and transferred onto PVDF membranes (Millipore, Burlington, MA, USA). Membranes were incubated with anti-HOXC6 (Santa Cruz, CA, USA, dilution, 1:1,000) and anti-β-actin (Santa Cruz, CA, USA, dilution, 1:1,000) overnight at 4 °C followed by incubation with a secondary antibody. Protein bands were visualized using ECL chemiluminescence (Millipore, Billerica, MA, USA) according to the manufacturer's recommendation.

## Cell proliferation assay

Cell proliferation was performed using Carboxyfluorescein Diacetate Succinimidyl Ester (CFSE) flow cytometry analysis on a BD FACSVerse™ (BD Biosciences, Frankin Lakes, NJ, USA). A final concentration of 5 μM CFSE was added to a single cell suspension of $10 \times 10^6$ cells/ml. The dye-cell suspension was then incubated in a 37 °C water bath for 10–15 min and then analyzed by flow cytometry. Cell proliferation was also assessed using the IncuCyte Live Cell Imaging System. Cells were plated in IncuCyte ImageLock 96-well plates (Essen BioScience, USA) at $1 \times 10^4$ cells per well. Cell confluency was monitored using the IncuCyte Live Cell Imaging System (Essen Bioscience, Ann Arbor, MI, USA) every 2 h.

## Cell migration and invasion assay

Transfected cells were plated in IncuCyte ImageLock 96-well plates (Essen BioScience, Ann Arbor, MI, USA). After cells adhered to the wells, an approximately 600-μm wide scratch was made in the cell layer using a WoundMaker (Essen Bioscience, Ann Arbor, MI, USA). The cells were then washed twice with PBS to remove floating cells. To assess migration, 100 μl of growth medium (with 2% FBS) was added to the cells. For invasion, the plate was coated with 30% Matrigel for 30 min at 37 °C, with an additional 50 μl of normal growth medium overlaid on to the plates. Wound closure was monitored using the IncuCyte Live Cell Imaging System (Essen Bioscience, Ann Arbor, MI, USA) every 2 h. Data analysis was conducted using Incucyte 2016A software.

## Construction of Sequence libraries

Total RNA was extracted from NSCLC cells A549 and PC9. Dynabeads Oligo (dT) 25 beads were used for isolating mRNA. RNA libraries were constructed with NEBNext® Ultra™ Directional RNA Library Prep Kit for Illumina® (NEB, Ipswich, MA, USA) with 3 independent replicates following the manufacturer's instructions. Paired-end sequencing (150 bp) was performing using Illumina HiSeq Xten. Raw data was uploaded to the GEO database (GSE121896).

## Bioinformatics analysis

We used HISAT2 (version 2.1.0) (*Kim, Langmead & Salzberg, 2015*) with default parameters to map the RNA-seq data to the GRCh37.p13 genome from GENCODE (*Harrow et al., 2012*). We aggregated the read counts at the gene level using HTseq (*Anders, Pyl & Huber, 2015*) and called differentially expressed genes (DEGs) with R package DESeq2 (*Love, Huber & Anders, 2014*). Genes were considered significantly differentially expressed when the Log2 fold change was >1 or <−1, and the adjusted *P* value was <0.05. Gene Ontology

analysis of the DEGs was performed using the R package ClusterProfiler (*Yu et al., 2012*) with $P < 0.05$.

## Statistical analysis

All of the statistical analyses were carried out using GraphPad Prism 7 and SPSS 21.0. Numerical data are presented as mean ± standard error, and were calculated using two-tailed Student's *t* test and Chi-square test. Two-way analysis of variance (ANOVA) was used to assess the statistical significance of multiple continuous variables. $P < 0.05$ was considered statistically significant.

# RESULTS

## High expression of *HOXC6* in Human NSCLC tissues

We examined the expression level of *HOXC6* in lung cancer using *in silico* approaches. Analysis of publicly available data revealed that there was a 5-fold and 8.8-fold increase in the expression of *HOXC6* in 2 independent lung cancer datasets, GSE30219 and GSE27262, respectively (Figs. 1A and 1B). To validate these findings, we used qRT-PCR to determine *HOXC6* expression in tumor samples from NSCLC patients. As shown in Fig. 1C, in 30 pairs of NSCLC tissue samples and adjacent non-tumor tissue samples, 66.6% (20/30) had elevated levels of *HOXC6* in comparison to their adjacent normal counterparts ($P = 0.01$). IHC analysis was consistent with the gene expression data, as HOXC6 protein was elevated in NSCLC tumor samples in comparison to adjacent normal controls (Figs. 1D and 1E). These results indicate that *HOXC6* is highly expressed in NSCLC tissues.

## The effect of *HOXC6* overexpression on the proliferation of NSCLC cells

To further explore the biological function of *HOXC6*, we used lentiviral vectors to establish *HOXC6*- overexpressing NSCLC cell lines. The resulting cell lines were named A549-HOXC6 and PC9-HOXC6. Control cell lines transfected with the empty vector were named A549-NEO and PC9-NEO. Transfection efficiency was then assessed by qRT-PCR and Western blot (Figs. 2A and 2B). We next examined the effect of *HOXC6* on the growth of NSCLC cells, by using an image-based proliferation assay. As shown in Fig. 2C, the proliferation rate was significantly increased in *HOXC6*- overexpressing A549 and PC9 cells in comparison to control cells. We also utilized the CFSE assay as a secondary proliferation assay to validate this observation. There was an obvious shift in the fluorescence signal in A549-HOXC6 and PC9-HOXC6 cells 48 h after CFSE labeling, indicating that *HOXC6* promoted the proliferation of both cell lines (Fig. 2D). These results demonstrate that overexpression of *HOXC6* increased the proliferation of NSCLC cells.

## The promotion of migration and invasion by *HOXC6* in NSCLC cells

Migration and invasion are recognized as important hallmarks of cancer (*Hanahan & Weinberg, 2011*). To investigate the effect of *HOXC6* on the migration and invasion of NSCLC cell lines, we made use of the wound healing and Matrigel assays. We observed significantly faster motility in A549-HOXC6 and PC9-HOXC6 cells in comparison to A549-NEO and PC9-NEO control cells in the wound healing experiment (Figs. 3A and

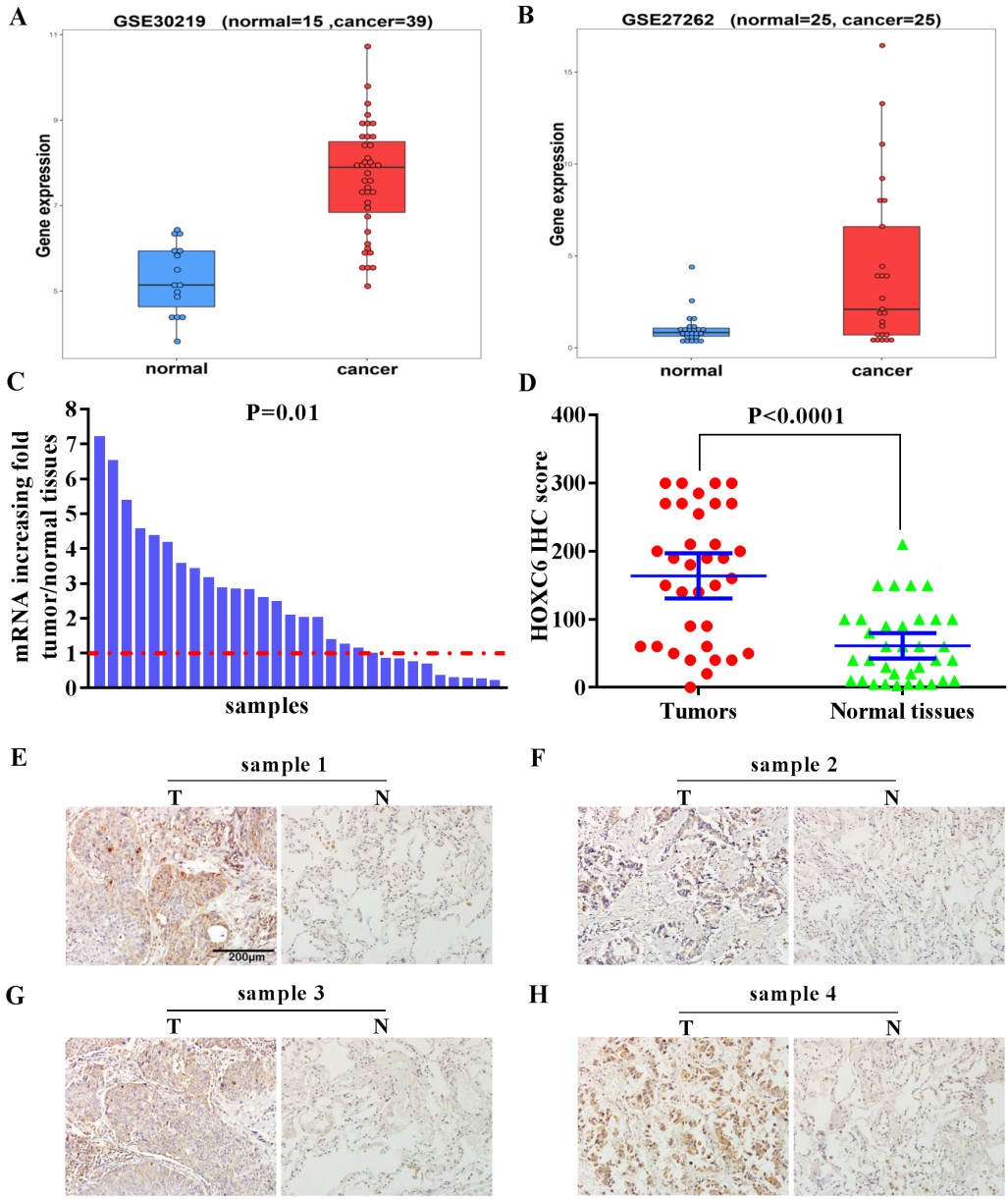

**Figure 1** **Elevated expression of *HOXC6* in NSCLC in comparison to adjacent normal tissues.** (A, B) The bioinformatic analysis of publicly available gene expression data sets for *HOXC6* expression in lung cancer tissues. Microarray dataset GSE30219 (normal = 15, cancer = 39) and GSE27262 (normal = 25, cancer=25) were download from Gene Expression Omnibus(GEO). *HOXC6* expression distribution in (A) and (B), with a fold change of 5 (*q*. value = 1.4332e–07) and 8.8 (*q*. value = 0.0039),which were calculated by R package limma (*Ritchie et al., 2015*). (C) Relative expression of *HOXC6* mRNA in 30 NSCLC and paired adjacent normal tissues as determined by qRT-PCR. The data were assessed using Chi-square test. (D) Expression of HOXC6 protein as determined by IHC. The IHC score of HOXC6 was calculated as the staining intensity (0, 1, 2, or 3) × the staining extent (0–100%). We compared matched samples as in (D) using paired two-tailed Student's *t* test. (E) Representative images of IHC staining of HOXC6 in NSCLC tumor (T) and adjacent normal tissues (N), Scale bar = 200 μm.

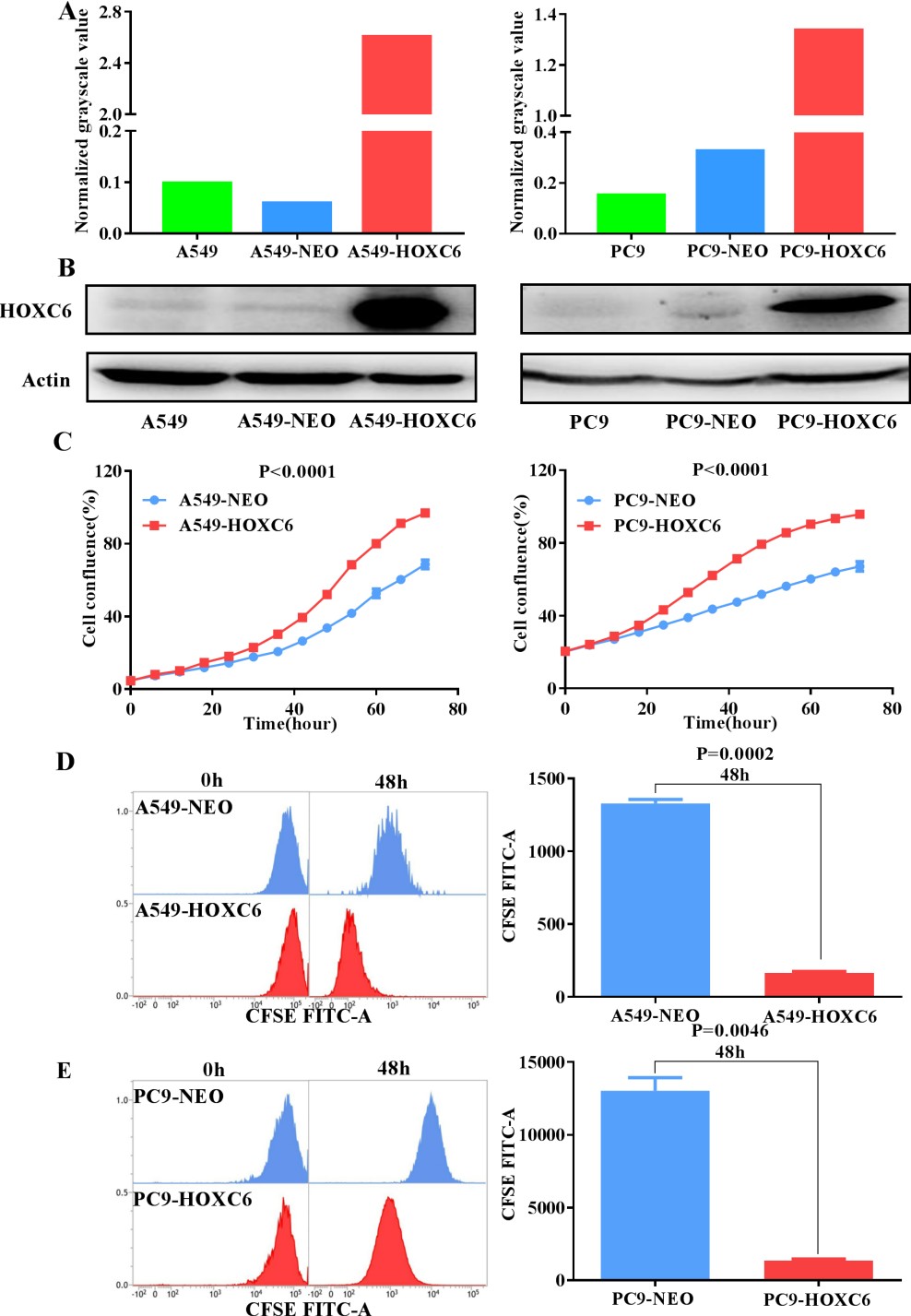

**Figure 2 Overexpression of *HOXC6* promotes NSCLC cell proliferation.** (A, B) Western Blot analysis of HOXC6 in A549 and PC9 *HOXC6*-expressing or control cells. (C) IncuCyte Live Cell Imaging System analysis of NSCLC cell lines A549 and PC9 transfected with HOXC6 or NEO lentiviral vectors. Cells (1 × $10^4$) were seeded in 96-well plates and monitored at 2-hour intervals, three replicates for each sample. (D) Flow cytometry analysis of CFSE-labeled *HOXC6*- expressing or control NSCLC cell lines, three replicates for each sample.

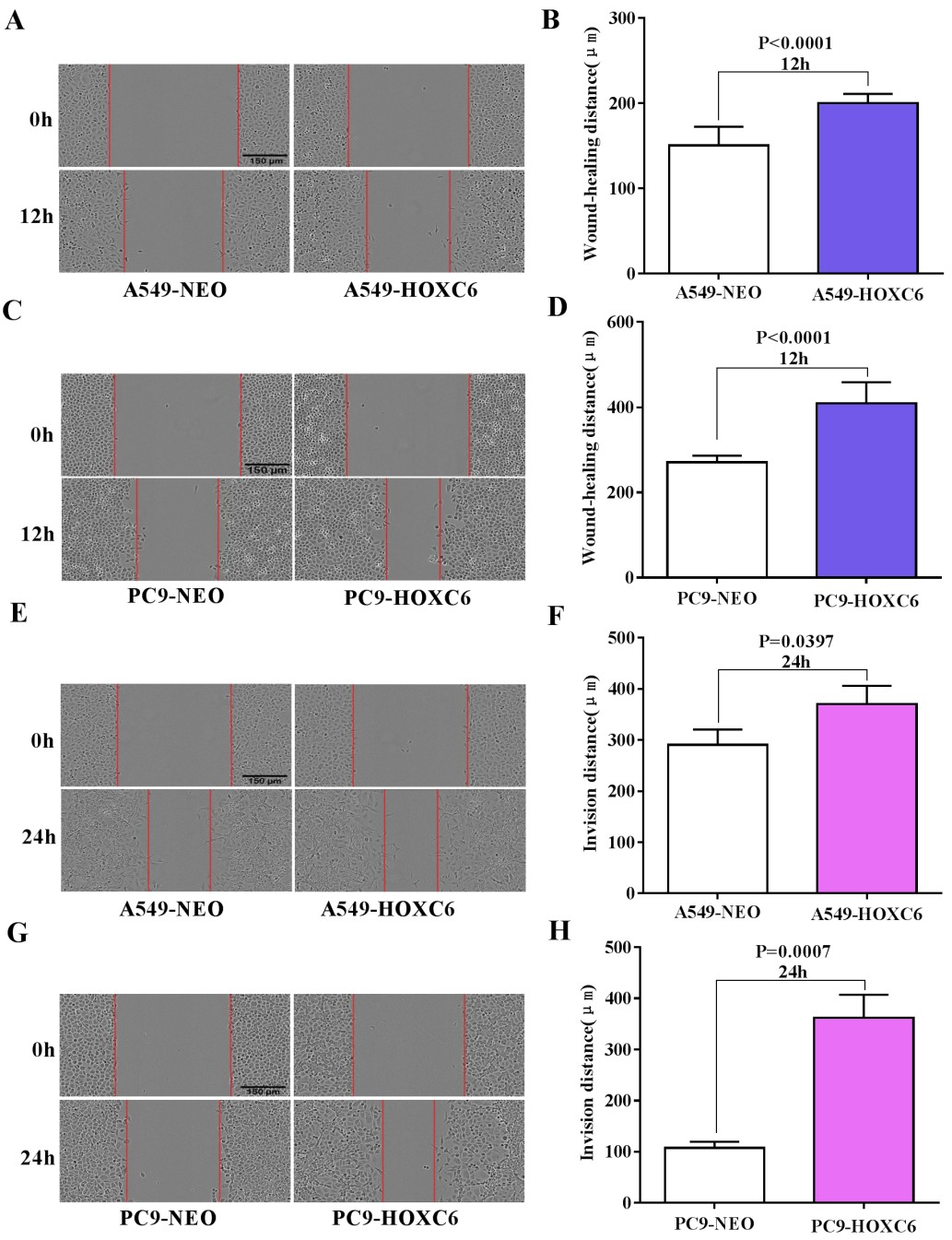

**Figure 3** **Overexpression of *HOXC6* promotes migration and invasion of NSCLC cells.** (A, B) Wound healing assay of A549 and PC9 cells expressing *HOXC6* or NEO. Scale bar = 150 μm. (C, D) Invasion of *HOXC6*-expressing or control A549 and PC9 cells was assessed in a wound healing assay using 30% Matrigel. Scale bar = 150 μm.

3B). Additionally, we observed that *HOXC6* increased the invasive capacity of NSCLC cells in Matrigel as shown in Figs. 3C and 3D. These results suggest that *HOXC6* may have a pro-tumorigenic role in NSCLC.

## The downstream targets of HOXC6 revealed by gene expression profiling in NSCLC cells

Just like other members of the homeobox superfamily, HOXC6 is a transcription factor that regulates the expression of several downstream target genes. RNA-seq analysis was performed to identify genes modulated by HOXC6 in NSCLC cells. The results (GSE121896) showed that there were 310 genes upregulated and 665 genes downregulated in A549-HOXC6 cells in comparison to A549-NEO control cells. In addition, we found that there were 112 genes upregulated and 385 genes downregulated in PC9-HOXC6 cells in comparison to PC9-NEO control cells (Fig. 4A). Gene Ontology analysis indicated that these genes modulated by HOXC6 may take part in various biological processes, such as morphogenesis and development, which is consistent with the previously reported functions of the homeobox gene superfamily (Fig. 4B). When we compared the genes regulated by HOXC6 in the two cell lines examined in this study, there were 9 common upregulated genes and 133 common downregulated genes (Fig. 4C). However, most of the genes whose expression levels were changed after HOXC6 transfection were different in these two cell lines. This unexpected result suggests that transcriptional regulation by HOXC6 is cell context-dependent.

## Pro-tumorigenic genes controlled by HOXC6 in NSCLC cells

Gene Ontology analysis did not clearly identify the molecular mechanisms underlying the pro-tumorigenic effect of *HOXC6*. Therefore, we supplemented this analysis by manually screening critical tumor-associated HOXC6-upregulated genes using functional analysis and literature review. Using this approach, we successfully identified several genes closely associated with tumor progression. *CEACAM6, SPARC, WNT6, CST1,* and *MMP2* were identified in A549-HOXC6 cells, while *CEACAM6* and *KRT13* were identified in PC9-HOXC6 cells. These genes have well documented roles in promoting cancer development (Table 1). We further analyzed the expression of these genes in various cancers through GEPAI (*Tang et al., 2017*). The results clearly demonstrate that the expression of these genes is associated with the progression of cancers (Fig. 5A), and that they are primarily involved in the regulation of cell proliferation, migration, and invasion. Furthermore, when we tried to analyze the function of the common upregulated genes, i.e., *HOXC6, SGK1, S100A4, MALL, CEACAM6, SLCO4A1-AS1, C11orf86, ENSG00000268621, and ENSG00000129270,* at least 5 of them, i.e., *SGK1, S100A4, MALL, CEACAM6* and *SLCO4A1-AS1,* have been reported to be implicated in the malignant phenotype of various cancers, while the remaining genes have not been well studied (*Egeland et al., 2017*; *Liang et al., 2017*; *Rizeq, Zakaria & Ouhtit, 2018*; *Yu et al., 2018*). We also analyzed the common downregulated genes and found that many of these genes such as *ANKRD12*, *KIAA1551,* and *APC* encode proteins with tumor suppressive functions (*Bai et al., 2013*; *Cheng et al., 2017*; *Lv et al., 2019*). These data indicate that HOXC6 could exert its oncogenic function by both activation of oncogenes and downregulation of the expression of tumor suppressive genes.

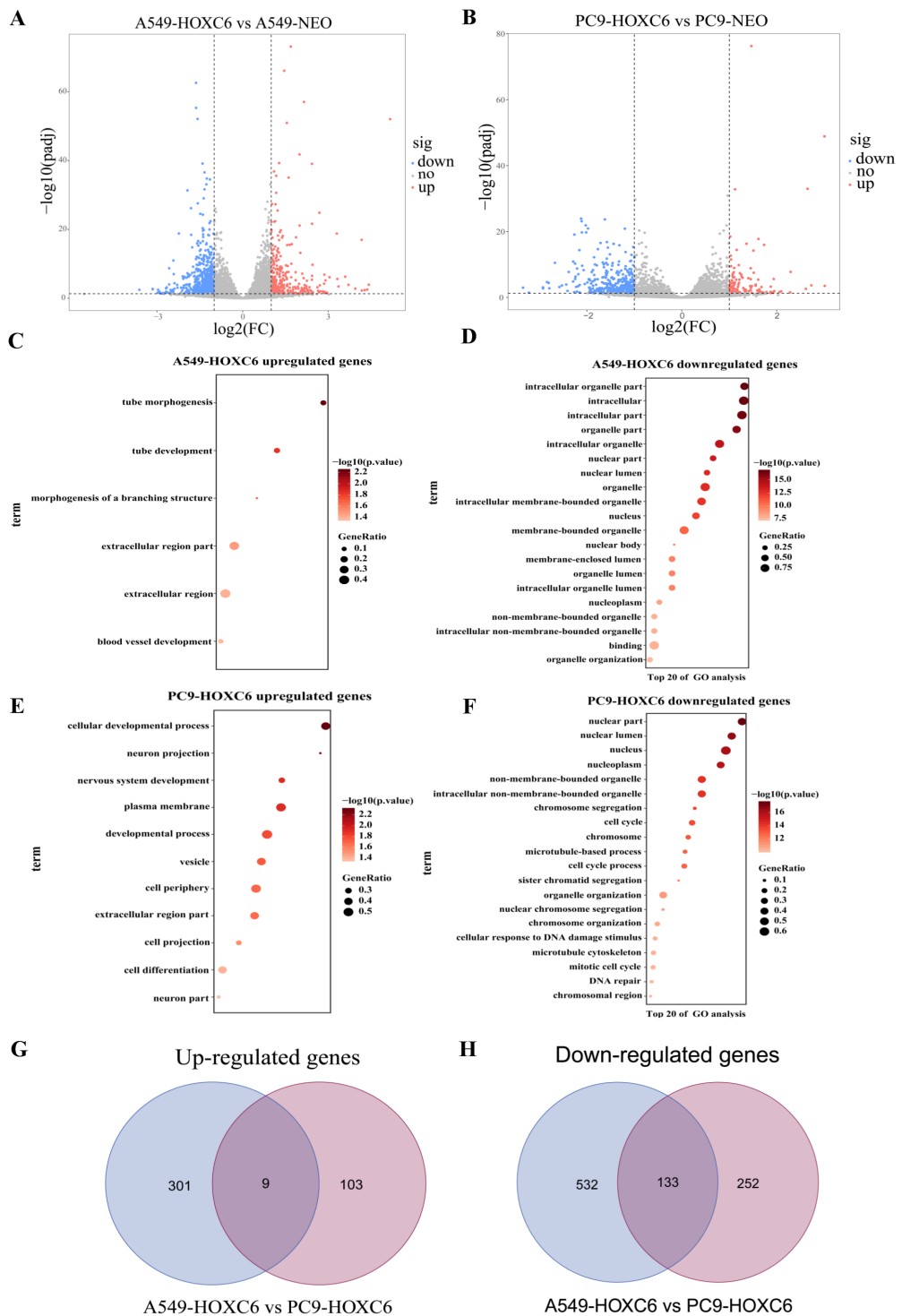

**Figure 4  Analysis of RNA-seq.** (A) The volcano map shows 310 upregulated genes and 665 downregulated genes in A549-HOXC6 cells in comparison to A549-NEO cells, and 112 upregulated genes and 385 downregulated genes in PC9-HOXC6 cells in comparison to PC9-NEO cells. Log2 fold change (Log2FC) >1 or <−1, and $P < 0.05$ were considered to be statistically significant. (B) Gene ontology (GO) analysis identified biological processes impacted by HOXC6-regulated genes in NSCLC cells. (C) Overlap of upregulated and downregulated genes in A549-HOXC6 and PC9-HOXC6 cells.

**Table 1  Potential target genes of HOXC6 in NSCLC.**

| Gene symbol | Gene description | Gene function[a] | References |
|---|---|---|---|
| CEACAM6 | Tumor marker | A, M, I, P, T | Rizeq, Zakaria & Ouhtit (2018) Zhang et al. (2013) |
| SPARC | Matrix-associated protein | P, M, CC, SF | Chang et al. (2018) Yusuf et al. (2014) |
| WNT6 | A family of highly conserved developmental control genes | P, M, A, CC, T | Yuan et al. (2013) Zheng & Yu (2018) |
| CST1 | The cystatin superfamily | P, M, I | Choi et al. (2009) Dai et al. (2017) |
| MMP2 | The major structural component of basement membranes | M, I | Kalhori & Törnquist (2015) Kuo et al. (2014) |
| KRT13 | Encoded a member of the keratin gene family | M, I, CC, A | Man et al. (2014) Li et al. (2016) |

**Notes.**

[a] Proliferation (P), Migration (M), Invasion (I), Angiogenesis (A), Cell cycle (CC), Tolerance (T), Stroma formation (SF).

# DISCUSSION

Lung cancer is the most frequently occurring malignancy worldwide. Due to the lack of effective measures for early diagnosis and treatment, the 5-year overall survival rate for NSCLC patients is only 16%–18% (*Siegel, Miller & Jemal, 2018*). The development of NSCLC involves both environmental and genetic changes, and the activation of oncogenes is also an important factor. Therefore, the identification of novel biomarkers is critical for the improvement of clinical outcomes for NSCLC patients.

In recent years, it has been found that the *HOX* gene is closely associated with the development and prognosis of various cancers. HOXC6 is a member of the HOX family that acts as a transcription factor and participates in the regulation of a number of genes during development. *HOXC6* has also been found to be highly expressed in several cancers. In this report, we provide several lines of evidence demonstrating that HOXC6 plays an oncogenic role in human NSCLC. First, *HOXC6* expression was shown to be elevated in NSCLC tissues. Second, *HOXC6* promoted the proliferation of NSCLC cells. Third, *HOXC6* enhanced the migration and invasion of NSCLC cells, which are fundamental hallmarks of cancer. Fourth, HOXC6 can upregulate the expression of genes critical for the development and progression of various cancers. Therefore, our work has established the basis for further investigation of *HOXC6* and its oncogenic roles in NSCLC.

To gain insight into the molecular mechanisms underlying the pro-tumorigenic functions of *HOXC6*, RNA-seq was performed to identify its downstream targets. A number of genes have been found to be modulated by HOXC6, but interestingly, we observed that there was very little overlap in the genes modulated by HOXC6 in two NSCLC cell lines. This result suggests that there is a cell context-dependent mechanism underlying the function of *HOXC6*. Since most transcription factors need to form complexes to specifically regulate their target genes, the cell context-specificity of HOXC6 may be caused by binding to different cofactors in different cell lines. Despite the fact

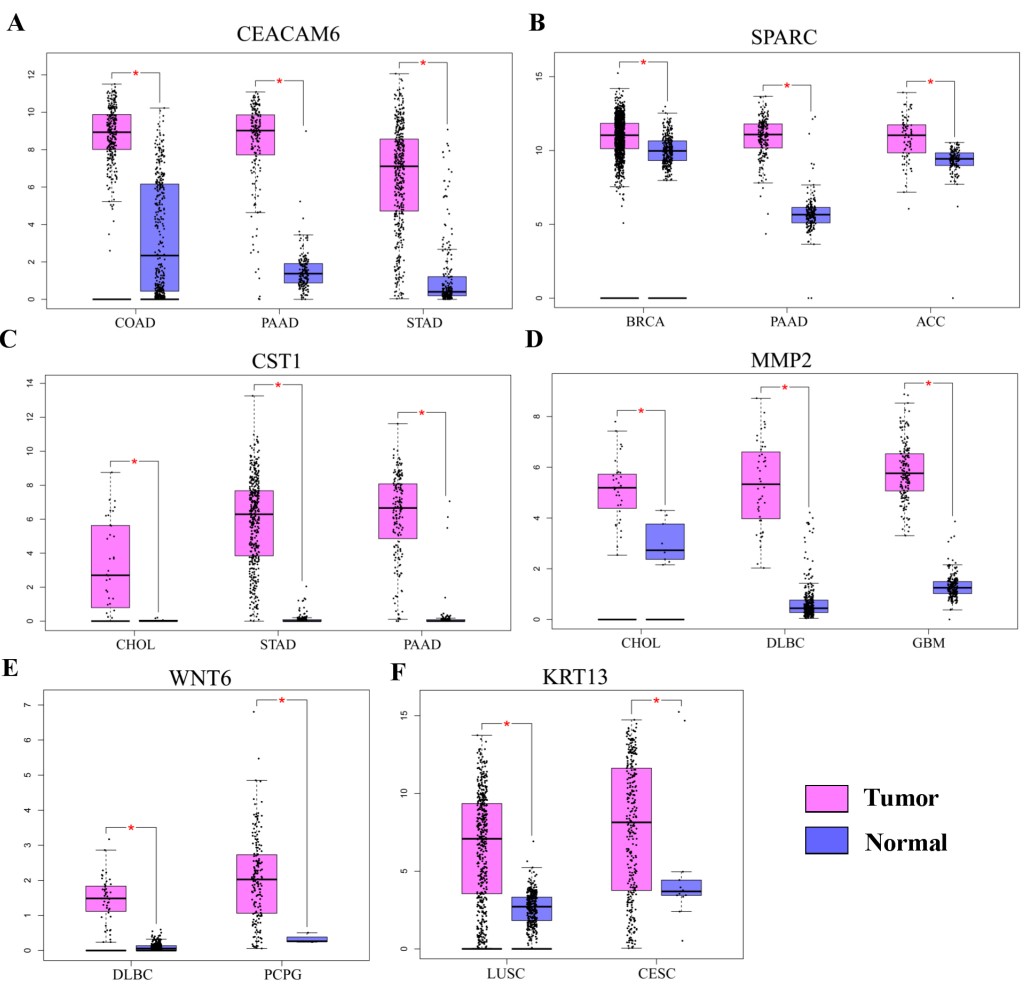

**Figure 5** **The expression of pro-tumorigenic genes in various cancers.** (A) The expression of *CEACAM6*, *SPARC*, *WNT6*, *CST1*, *MMP2*, and *KRT13* in various cancers, $^*P < 0.05$. Breast invasive carcinoma (BRCA), pancreatic adenocarcinoma (PAAD), adrenocortical carcinoma (ACC), cholangiocarcinoma (CHOL), stomach adenocarcinoma (STAD), lymphoid neoplasm diffuse large B-cell lymphoma (DLBC), pheochromocytoma and paraganglioma (PCPG), colon adenocarcinoma (COAD), lung squamous cell carcinoma (LUSC), cervical squamous cell carcinoma and endocervical adenocarcinoma (CESC).

that different sets of genes are regulated by HOXC6 in various cell lines, transfection of *HOXC6* into both cell lines used in this study generated similar phenotypic effects. Consistent with this result, we found that HOXC6 is a master regulator of many genes, which have documented pro-tumorigenic functions. In our studies, we identified several genes, including *CEACAM6, SPARC, WNT6, CST1, MMP2,* and *KRT13*, which have been extensively studied and demonstrated to be involved in the regulation of tumor growth, migration and invasion, cell cycle, and apoptosis (*Chang et al., 2018*; *Choi et al., 2009*; *Dai et al., 2017*; *Kuo et al., 2014*; *Rizeq, Zakaria & Ouhtit, 2018*; *Wang et al., 2018*; *Yusuf et al., 2014*; *Zheng & Yu, 2018*). In other types of cancers, HOXC6 has also been shown to have the capacity to regulate these cancer-related genes. For example, HOXC6 directly

regulates gene expression of Bone morphogenetic protein 7 (*BMP7*), Fibroblast growth factor receptor 2 *(FGF2)*, and Platelet-derived growth factor receptor *(PDGFR)* in prostate cancer (*McCabe et al., 2008*). HOXC6 promotes the migration, invasion, and progression of gastric cancer by upregulating Matrix metalloproteinase 9 *(MMP9)* (*Chen et al., 2016a*). These results suggest that HOXC6 can promote tumorigenesis by regulating distinct sets of genes in various cellular contexts.

## CONCLUSION

In conclusion, we demonstrate that *HOXC6* was highly expressed in NSCLC tissues and correlated with the malignant phenotype of NSCLC cells. In addition, bioinformatics analyses showed that HOXC6 may enhance lung cancer progression by regulating the expression of pro-tumorigenic genes involved in proliferation, migration, and invasion. Our study highlighted the oncogenic potential of *HOXC6* and suggests that it is a candidate molecular marker for the diagnosis and treatment of NSCLC.

### Funding

This work was supported by the joint program on the science and technology collaboration of Southwest Medical University and the government of Luzhou city (2018LZXNYD-PT04), the Science Technology Support Plan Projects of Luzhou (No. 2015LZCYD-S02) and the Project Program of The Affiliated Hospital of Southwest Medical University (17160). The funders had no role in study design, data collection and analysis, decision to publish, or preparation of the manuscript.

### Grant Disclosures

The following grant information was disclosed by the authors:
Southwest Medical University.
Government of Luzhou city: 2018LZXNYD-PT04.
Science Technology Support Plan Projects of Luzhou: 2015LZCYD-S02.
Affiliated Hospital of Southwest Medical University: 17160.

### Competing Interests

The authors declare there are no competing interests.

### Author Contributions

- Yingcheng Yang conceived and designed the experiments, performed the experiments, analyzed the data, prepared figures and/or tables, authored or reviewed drafts of the paper, approved the final draft.
- Xiaoping Tang conceived and designed the experiments, performed the experiments, analyzed the data.
- Xueqin Song, Li Tang, Yong Cao, Xu Liu, Xiaoyan Wang, Yan Li and Minglan Yu performed the experiments.
- Feng Chen contributed reagents/materials/analysis tools.
- Haisu Wan conceived and designed the experiments, analyzed the data, contributed reagents/materials/analysis tools, authored or reviewed drafts of the paper, approved the final draft.

## Human Ethics

The following information was supplied relating to ethical approvals (i.e., approving body and any reference numbers):

The Affiliated Hospital of Southwest Medical University granted Ethical approval (k2018003-r) to carry out the study within its facilities.

## Data Availability

GEO accession number GSE121896.

GEPAI analysis can be found at http://gepia.cancer-pku.cn/index.html and entering CEACAM6, SPARC, WNT6, CST1, MMP2, or KRT13.

## Supplemental Information

Supplemental information for this article can be found online at http://dx.doi.org/10.7717/peerj.6629#supplemental-information.

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
