# Peer review of "Evidence for an oncogenic role of HOXC6 in human non-small cell lung cancer"

_PeerJ, doi:10.7717/peerj.6629_

## Round 0.1 · original submission · Major Revisions

Please revise the MS according to the reviewers' suggestions

Reviewer 1 ·

Basic reporting

The authors do a commendable job showing HOXC6 as a potential biomarker for NSCLC. The manuscript is well written and would be easily assessable to the international audience. The authors did a commendable job in introducing the topic and also giving a comprehensive background. Figures are relevant with proper labeling, legend and description.

Experimental design

The research is thorough and is well thought. I would also suggest that the authors show the proliferation of the native A549 and PC9 cell lines on its own and compare Neo and HOXC6 to it in Figure 1A and 1B. This would make sure, that proliferation rate is not changing by the introduction of the vector backbone.

Validity of the findings

The data is robust and proper statistical methods were used. The conclusions for the most part are well stated.
I have certain reservations about the scratch assay that the authors have performed. First, 2% FBS is more than double of what is usually used and I would suggest blocking the mitosis a few hours earlier. For instance, the authors can add Cytosine b-d-arabinofuranoside hydrochloride, a selective DNA synthesis inhibitor but does not inhibit RNA synthesis. The other reservation to this is since both the cell lines are highly proliferative; it raises questions about actual migration. I would suggest repeating this assay incorporating both these changes in the experiment.

Additional comments

The manuscript is well written and is easy to follow. It has a concise methodology and results are well presented and adhere to the standards of Life sciences. However, there are some issues that need to be clarified, a few experiments performed before acceptance. Please see my further comments below:-
1] The second result topic (196 – 203) is already been explained in the Methods section and is just a duplication.

2] Can you name those 9 common upregulated genes and mention their role in tumor survival and progression. Also, are there any common downregulated genes?

·

Basic reporting

In this manuscript, Yingcheng Yang, et al. aims to investigate the role of HOXC6 in non-small cell lung cancer (NSCLC). They show that HOXC6 is highly expressed in NSCLC samples, and that overexpression of HOXC6 leads to cell proliferation, migration, and invasion. The oncogenic role of HOXC6 has already been characterized in many cancers (Hamid, Agus Rizal AH, et al. "The role of HOXC6 in prostate cancer development.”; Grier, D. G., et al. "The pathophysiology of HOX genes and their role in cancer.”; Shang, Jianhong, et al. "HOXC6 expression is associated with a poor prognosis in early-stage cervical squamous cell carcinoma.”), and the conclusion from this study is consistent with the previous findings. While the overall analyses on cell lines are comprehensive and the writing is generally clear, several major points need to be addressed.

Experimental design

The experiments are well-designed, yet some improvements need to be made. Please see my general comments for the authors.

Validity of the findings

The statistical analysis needs to be improved. Please see my general comments for the authors.

Additional comments

General concerns:
1. The title is confusing and misleading. First, the authors do not indicate what kind of “molecular marker” HOX6 can be used for, i.e., is it considered a molecular marker for NSCLC diagnosis or prognosis? Second, the data from the manuscript support the oncogenic potential of HOXC6; however, this does not indicate HOXC6 can be used as a molecular marker if sensitivity and specificity of this marker have not been evaluated in this study.
2. Validity of cell lines. Two NSCLC cell lines were used in this study: A549 and PC9. In Figure 1, the authors show that at both transcription and translation levels, HOXC6 is highly expressed in NSCLC tissue. However, in Figure 2, according to the western blot, both cell lines do not express HOXC6 when transfected with an empty vector. Therefore, it is questionable whether these two cell lines are suitable for evaluating the oncogenic effect of HOXC6.

Specific comments to the manuscript:
1. In Figure 1A and 1B, please indicate the sample size. If the error bar indicates standard error as stated in the statistical analysis section, the standard deviation must be huge and it will not be suitable to compare between groups using student t test. A nonparametric test needs to be considered. When comparing matched samples as in Figure 1C and 1D, please indicate which t test (paired or independent) has been applied. Please indicate the stage of the clinical NSCLC tissue, as it is likely that the expression level of HOXC6 may correlate with the stage of the cancer. Also, please make the scale bars in E big enough to read.
2. In Figure 2, please indicate the quantification of the western blot data.
3. In Figure 3, please indicate the sample size in the figure legend. Also, since the proliferation of the cells is significantly changed upon HOXC6 expression, is there any change in cell cycle regulation? The authors may need to provide a cell cycle analysis by flow cytometry and/or include the expression level of some cell cycle regulators.
4. In Figure 4, please make sure the quantification is accurate. How was the migration distance calculated? In panels A and B, with HOXC6 overexpression, the migration distances seem to be at least twice as the control according to the red lines in the images, which is inconsistent with the quantifications. Also, please indicate scale bars in the images.
5. In Figure 5, it is surprising to see that only 9 upregulated genes overlap between the two cell lines. Do these cell lines exhibit difference in proliferation, migration, and invasion without HOX6 overexpression? This can be a supplementary information but is important to elucidate the role of HOXC6. The regulatory role of HOXC6 on transcription may be dependent on the proliferative capacity of the cells, and its downstream gene targets are subjected to other feedback regulatory mechanisms.
6. In Figure 6, the authors identified several genes that are significantly upregulated in various cancer tissue. Indeed, all these genes, including WNT6, MMP2, CEACAM6 have been well-acknowledged to promote tumorigenesis. However, this does not mean these genes are “controlled by HOXC6 in NSCLC cells”. The overexpression of HOXC6 changes the whole transcriptome profile of the cancer cells, but no clue can be provided regarding whether HOXC6 is upstream or downstream.

---

## Round 0.2 · Major Revisions

Please follow the suggestions of Reviewer 2.

Reviewer 1 ·

Basic reporting

Its good

Experimental design

The required experiments were carried out

Validity of the findings

It makes sense after the revision

·

Basic reporting

I would like to thank the authors for the revision. The quality of the manuscript has been improved. However, there are still some issues need to be addressed.

Experimental design

No comment

Validity of the findings

No comment

Additional comments

1. There are still many grammar mistakes in this version. I would suggest the authors do a careful correction or ask a native speaker for proofreading. Here are some examples:
1) We comparing (compared) matched samples as in Fig. 1D using paired two-tailed Student’s t test.
2) In conclusion, we demonstrate that HOXC6 was highly expression (expressed) in NSCLC tissues and correlated with the malignant phenotype of NSCLC cells.
3) Prompt (“prompt” is rarely used in this way), HOXC6 is a potential molecular marker for the diagnosis and treatment of NSCLC.
2. I would suggest the authors not use standard error (SEM) in the figures as this does not reveal the true data variation, especially when the sample size is large (for example, Figure 1D). SD or 95%CI will be better.
3. I suggest the authors include Figure C in the rebuttal letter as a supplementary Figure for the manuscript. It is important that the expression of HOX6 be justified in these two cell lines. It would be unconvincing if the authors claim that HOX6 is highly expressed in NSCLC, but the two cell lines they used for all the in vitro experiment have no endogenous expression of HOX6 shown in Figure 2B.
4. In Figure 3, the quantification in A does not agree with the image presented. In the rebuttal letter, the authors argue that “The quantification result is derived from the statistical analysis of multiple wells and the image shown here is taken from one of these wells.” I would suggest the authors 1) choose a representative image for the figure; and 2) provide details of the quantification in the method section (e.g. how the “wound healing distance” was defined).

---

## Round 0.3 · accepted · Accept

Thank you for your contribution.

# ·

Basic reporting

No comment

Experimental design

No comment

Validity of the findings

No comment

Additional comments

I appreciate the author's responses to my comments. All my concerns have been addressed in the revised manuscript. I think that with the added clarifications, this manuscript is now acceptable for publication in PeerJ.